# Hutchinson-Gilford Progeria Syndrome—Current Status and Prospects for Gene Therapy Treatment

**DOI:** 10.3390/cells8020088

**Published:** 2019-01-25

**Authors:** Katarzyna Piekarowicz, Magdalena Machowska, Volha Dzianisava, Ryszard Rzepecki

**Affiliations:** Laboratory of Nuclear Proteins, Faculty of Biotechnology, University of Wroclaw, Fryderyka Joliot-Curie 14a, 50-383 Wroclaw, Poland; katarzyna.piekarowicz@uwr.edu.pl (K.P.); magdalena.machowska@uwr.edu.pl (M.M.); volha.dzianisava2@uwr.edu.pl (V.D.)

**Keywords:** HGPS, laminopathy, lamin A/C, progerin, gene therapy, miR9

## Abstract

Hutchinson-Gilford progeria syndrome (HGPS) is one of the most severe disorders among laminopathies—a heterogeneous group of genetic diseases with a molecular background based on mutations in the *LMNA* gene and genes coding for interacting proteins. HGPS is characterized by the presence of aging-associated symptoms, including lack of subcutaneous fat, alopecia, swollen veins, growth retardation, age spots, joint contractures, osteoporosis, cardiovascular pathology, and death due to heart attacks and strokes in childhood. *LMNA* codes for two major, alternatively spliced transcripts, give rise to lamin A and lamin C proteins. Mutations in the *LMNA* gene alone, depending on the nature and location, may result in the expression of abnormal protein or loss of protein expression and cause at least 11 disease phenotypes, differing in severity and affected tissue. *LMNA* gene-related HGPS is caused by a single mutation in the *LMNA* gene in exon 11. The mutation c.1824C > T results in activation of the cryptic donor splice site, which leads to the synthesis of progerin protein lacking 50 amino acids. The accumulation of progerin is the reason for appearance of the phenotype. In this review, we discuss current knowledge on the molecular mechanisms underlying the development of HGPS and provide a critical analysis of current research trends in this field. We also discuss the mouse models available so far, the current status of treatment of the disease, and future prospects for the development of efficient therapies, including gene therapy for HGPS.

## 1. Introduction

Hutchison-Gilford progeria syndrome (HGPS; OMIM 176670) is a rare human genetic disorder linked with a subset of specific mutations in the *LMNA* gene, coding for lamin A and lamin C proteins. The *LMNA* gene is located at position 1q22. Interestingly, different sets of mutations in the *LMNA* gene and genes coding for interacting proteins, such as emerin (*EMD/STA* gene) and BAF (barrier-to-autointegration, *BANF1* gene), give rise to a variety of genetic disorders collectively called laminopathies [1,2,3]. It is currently thought that at least 11 distinct disease phenotypes can be defined within the laminopathy group. These include: EDMD1 (Emery-Dreifuss muscular dystrophy 1, OMIM 310300), EDMD2 (OMIM 181350), EDMD3 (OMIM 616516), DCM (dilated cardiomyopathy, OMIM 115200), FPLD2 (Dunnigan familial partial lipodystrophy type 2, OMIM 151660), CMT2B1 (Charcot–Marie–Tooth disorder, type 2B1, OMIM 605588), heart-hand syndrome, Slovenian type (OMIM 610140), Malouf syndrome (OMIM 212112), MADA (mandibuloacral dysplasia with type A lipodystrophy, OMIM 248370), and RD (restrictive dermopathy, OMIM 275210). MADA is a type of mandibuloacral dysplasia associated with mutation in the *LMNA* gene, while MADB is associated with *ZMPSTE24* gene coding for cysteine proteinase (prenyl protease 1 homolog), which among other functions, is responsible for maturation of prelamin A by cleaving off the farnesylated C-terminus. Both are also considered as progeroid laminopathies. Each disorder from the laminopathy group has its own unique phenotype and, typically, a set of common phenotypes with other diseases. Some of the *LMNA* mutations give rise to phenotypes that may be classified into two or more separate disorders. Mutations of arginine 527 such as R527C, R527H, and R527P may be asymptomatic, progeric, result in MADA (with or without myopathy) or cause EDMD2 alone or combined with FPLD2 [4,5,6] (www.umd.be/LMNA/). Moreover, the particular phenotype of the particular mutation can be modified/affected/masked by the genetic background of the patient [7].

Similar genetic disorders to HGPS, with at least partially similar genetic background and molecular mechanisms of pathogenesis, have been recently characterized. Nestor-Guillermo progeria syndrome (OMIM 614008) [8,9] arises due to mutations in the *BANF1* gene (11q13.1) coding for BAF protein, which is an interacting partner for, among others, emerin and lamin A/C complexes with chromatin. RD is an autosomal recessive, lethal disorder associated with mutations in two genes: *LMNA* and *ZMPSTE24* [10,11].

## 2. Phenotype and Genetic Background

The phenotype of the HGPS is variable [12]. Typical childhood-onset phenotype includes postnatal growth retardation, midface hypoplasia, micrognathia, osteoporosis, absence of subcutaneous fat, low body weight, lipodystrophy, decreased joint mobility, alopecia, and premature aging. Median life expectancy is about 13 years. The major direct causes of death are cardiovascular problems [13]. Classical HGPS has only autosomal dominant mode of inheritance and a clearly defined molecular background. The progeria-related phenotypes associated with so-called non-classical mutations are frequently described as progeroid laminopathies, atypical progeroid syndromes, or MADA [4,5]. They are autosomal dominant or recessive.

For progeroid laminopathies the time of onset of the disease, set of symptoms and severity depend on the type of mutations [14,15,16,17,18]. The vast majority of autosomal dominant type of the progeric laminopathies arise from the so-called “classical” mutation in the *LMNA* gene—this mutation causes HGPS [19,20]. It is mostly a de novo single nucleotide substitution mutation c.1824C > T in exon 11 which should be “silent” since both nucleotide triplets (wt and mutant) code for glycine (p.G608G mutation). Unfortunately, such a single nucleotide change activates the cryptic donor splicing site for lamin A-specific transcript processing only (transcript variant 7) (according to NCBI database; www.ncbi.nlm.nih.gov). The splicing for lamin C transcript remains unaffected (see Figure 1 for details). The mutation-activated new splicing site leads to synthesis of a transcript with part of exon 11 missing and results in synthesis of mutant lamin A, which is called progerin. Progerin lacks 50 amino acid residues encoded by the missing exon 11 fragment. This deletion removes, among others, a target site for ZMPSTE24 cysteine proteinase which is involved in processing and maturation of prelamin A protein. In normal cells, this protease cleaves off the farnesylated (on cysteine residue) CAAX motif, located on the C-terminal end of prelamin A. The absence of the ZMPSTE24 protease cleavage site in progerin results in persistence of constantly prenylated progerin. This in turn directs progerin preferentially to the nuclear lamina/nuclear membranes. Heterozygous cells then have wt lamin A expression from the wt allele, lamin C expression from both alleles and progerin expression from the mutant allele. Surprisingly, it appears that, at least in some normal but “old cells” or fibroblasts from biopsies from elderly persons without the mutation, it is possible to detect spontaneous activation of this cryptic splicing donor site [21]. The activation extent increases with the cell passage number or person’s age. This discovery already stimulates a series of general questions. Is this process a cause or effect of aging? Both options are similarly intriguing, although dysregulation of general splicing regulation, which also affects *LMNA* splicing, seems to be a more acceptable hypothesis at this point since the original study has not been either reproduced or thoroughly studied. In the general view of aging, it is assumed that in HGPS and in Werner syndrome, the rate of brain aging is significantly lower than the rate of whole body aging, whereas in Down syndrome, the brain is aging faster than other organs due to brain-specific accumulation of amyloid-beta and chronic oxidative stress (for review see: [22]). In this view, the lamin A level (and progerin level) in tissues would be inversely proportional to aging susceptibility of tissues. It is surprising that this potential correlation has not been explored further.

It should be pointed out that two mutations in the *LMNA* gene result in another severe disorder called restrictive dermopathy (RD) [23,24,25,26,27]. RD is a lethal, early onset (frequently intrauterine) disorder resulting from a homozygous, inactivating mutation in the *ZMPSTE24* gene or a homozygous “classical” mutation in the *LMNA* gene. This connection immediately points to major molecular pathways responsible for development of the disease phenotype, which develops when progerin (permanently farnesylated, truncated lamin A) is expressed from both alleles. This also indicates the importance of the ratio between progerin and lamin A/C in the mechanisms of phenotype development.

The progeroid laminopathies (or MADA phenotypes) are associated with other mutations than “classical”. Mutations identified so far, beyond any doubts, are missense mutations affecting the head domain (P4R) coil 1B (e.g., A57P, L59R, E138K, L140R, S143F, E145K, or similar) [28,29], one spot in coil 2 and several spots in the tail region (R471C, R527C/H, K542N, S573L, G608S) (Figure 1) [14,16,17]. Most of such mutations result in a more complex phenotype which partly overlaps with other laminopathies such as FPLD2, DCM, EDMD2, or a mixture of them [18,30].

What is the current view of the molecular background of the disease phenotype development in HGPS progeria?

If we take into the consideration data reported from mice model studies and patients we might encounter certain discrepancies on the role of permanent farnezylation of progerin, progerin presence, and the ratio between progerin and lamins A/C in the development of the phenotype [31,32,33,34,35].

If we take the current view of the role of nuclear lamina structure and its components into consideration, we can divide all possible mechanisms into several, overlapping categories which, for the sake of simplicity of considerations, can be divided into the following: modified gene regulation due to remodeled chromatin structure [36], abnormal regulation of transcription via abnormal signaling pathway modulation [37,38,39,40,41,42], abnormal mechanical properties and reaction to external forces/ECM (extracellular matrix) environment [43,44,45,46,47,48].

The role of lamins, particularly lamin A/C, and interacting proteins in few of the processes mentioned above has already been discovered and recently reviewed in several publications [13,49,50,51,52,53,54]. Due to the complexity of interactions of lamins with signaling pathways and processes, it is difficult to point out only one or two major factors essential for the development of a particular phenotype. It is possible though, and achievable for the simplest disorders from the laminopathy group such as EDMD1 and EDMD2 [55,56,57,58] or FPLD2 [59,60]. Indeed, a few such pathways have been discovered, and a few possible treatment strategies have been suggested. For HGPS, it is not possible yet because we have to deal with not a simple, phenotypic null mutation, but a fully active protein with extra ability to associate with membranes (with unknown efficiency) and the possibility of relocating all possible interactions to the nuclear lamina [41,61,62]. The missing fragment of 50 amino acids from the C-terminus of progerin contains not only the target site for protease, but also has been suggested as a chromatin binding site [63] and encompasses several regulatory regions located in highly unstructured regions (serine and glycine rich) for phosphorylation by signaling kinases. Phosphosites and serine/glycine rich regions are evolutionarily conserved in vertebrate lamin A [54]. Due to the deletion, the progerin C-terminus amino acid residue sequence resembles the lamin B2 sequence. This, in turn, raises the question of whether progerin preferentially interacts with the B-type or A-type network.

We might assume that progerin presence changes not only the mechanical properties of the entire cell nucleus and chromatin [64,65], but also the chromatin structure and division into the functional transcriptionally active domains (TADs) and structural lamina associated domains (LADs), which in turn modify the gene regulatory mechanisms in differentiated tissue and most of all during development of tissues and organs. Since lamins and interacting proteins play a regulatory role modulating signaling pathways and act as a hub integrating different signaling pathways, we may assume that progerin presence affects major regulatory pathways. This includes stress response [66,67] and DNA repair [68], proliferation [69,70], the mitogen-activated protein kinase/Extracellular signal-regulated kinase (MAPK/ERK) system as well as the Wnt/β-catenin pathway, the Smad pathway, and Notch signaling [3,53,71], and the mTOR pathway [72]. Additionally, the direct link between chromatin and nuclear lamina structures with cytoskeleton and ECM should not be forgotten. This point is of special importance especially in stress-generating or stress-enduring tissues such as muscles, including cardiac muscle, the cardiovascular system, and particularly smooth muscle cells, where the progerin expression level is the highest or in the highest proportion compared to lamin A/C [73]. Since progerin is predisposed to be located at the nuclear lamina and nuclear envelope, it might relocate all the interactions in which it participates, including with chromatin complexes (and LADs), into the nuclear lamina or into the blebs specifically [74,75]. It should be pointed out that, in normal cells, a significant fraction of wt lamin A, specifically phosphorylated [54,76,77,78] is located in the nucleoplasm. A decrease of the intranuclear fraction of lamin A might reduce, for example, the intranuclear fraction of LAP2α and reduce the ability of S-phase entry or translocation of LADs into blebs [75], mechanically forcing chromatin reorganization/separation of lamin A LADs with lamin B1 LADs, since both form independent networks [36]. We might speculate that transcriptionally active lamin A-LADs are more prone to translocation to blebs [75]. On the other hand, previous studies reported that in HGPS cells in general, chromatin is converted into the heterochromatin-rich state and treatment reverses the blebbing and chromatin state [21,79,80,81] see also [82]. These controversies over chromatin state and transcription might be explained by differences in cellular models. Relocation of lamin A from lamin B was demonstrated only in progeroid laminopathy MADA patient fibroblasts with mutation S143F, while the reorganization of chromatin into the heterochromatin state was demonstrated in HGPS patient cells—namely “classical” mutation. Here arises another question regarding progerin and lamin skeletal structures. It is well documented that A-type lamins and B-type lamins form separate networks in mammalian cells [36] and fly cells [83], but what is the preferential assembly partner for progerin (if there is any)? Some reports suggested that progerin copolymerizes with both A-type lamins and B-type lamins [84]. This discovery would nicely explain the controversies discussed above. Lamin A S143F still behaves as lamin A as it forms, separately from lamin B, bleb structures. Progerin due to the permanent farnesylation and carboxy methylation (by isoprenylcysteine methyltransferase, ICMT) may interact with the lamin B network at the nuclear envelope and still retains preference for polymerization to lamin A. Apart from the C-terminal modifications and short deletion it is identical in sequence with lamin A. We will discuss potential consequences of the deletion further in following sections. In such a view the critical question is whether the treatment of HGPS model mice with ICMT inhibitors and phenotype improvement [85] results in the restoration of the separation of the lamin A/progerin network from the lamin B network or from increased autophagy of farnesylated and non-methylated progerin, or both.

Apart from the questions about possible mechanisms associated with different chromatin interactions discussed above and the “interactome” of progerin [86], the overall phenotypic effect might be stronger the higher the proportion of progerin to lamin A/C in a particular tissue is.

If we consider such simple relocation to the nuclear lamina of a particular, single LAD region we might predict that genes located in such a domain will be probably subjected to a different environment and be abnormally regulated (e.g., suppressed, activated or inactivated) since lamins are responsible not only for proper forparticular cell type, LADs formation, but through affecting LADs formation or not also affect TAD-TAD interactions and change gene expression profiles [87]. We might speculate that these are only a few or limited number of genes and that there is a fair chance that the sister chromosome TAD will be still active in the old way. There is a fair chance to be bound by wt lamin A. However, what if is not? What if we have, for example, 10 LADs relocated, or 100 [87]? On top of that we have chromosome territories and domain organization for chromosomes different in each cell or even TADs on parent chromosomes are differently organized. Thus, relocation of TAD from the father chromosome may give rise to a different expression pattern than relocation of the same TAD from the mother chromosome [88].

In the above line of thoughts, we are trying to demonstrate only that the simplest possible mechanism of pathogenesis, which is relocation of a particular TAD to the nuclear lamina, may give rise to many different outcomes for a particular cell in question. If we add to this stochastic process the modulatory activity of lamin A/progerin on signaling pathways, we will get too many possible pathological mechanisms to consider them independently and to deal with them independently and adequately. This should also be kept in mind when considering therapeutic strategies.

This means that design of an efficient therapy for HGPS cannot focus on particular pathways or processes, but would require progerin or prenylated mutant lamin A to be eliminated from patient cells as efficiently as possible. This distinction has been made just to acknowledge the treatment option based on the idea of efficient elimination of prenylation of progerin or specific elimination of progerin itself. The other lesson from our discussions might be the thesis that entire mechanisms associated with progeria should be studied in an animal model system when the presence of the mutation is subjected to different environmental conditions of particular tissues and long-term interactions being set and modified through the entire development. Therefore animal models and animal model studies are of extreme value to gain knowledge of the pathogenetic mechanisms of the HGPS and progeric laminopathies.

## 3. Mouse Models for HGPS

A number of mouse models for progeria have already been developed to investigate the molecular mechanism and potential treatment. They differ in genetic background and observed phenotype, but changes in nuclear shape are commonly observed.

The first generated models were based on ZMPSTE24 deficiency. Pendas and colleagues replaced exons two–three in the *ZMPSTE24* gene with the lacZ cassette, creating *ZMPSTE24*-null mice. Changes in murine phenotype were observed only for homozygotes. No lamin A or prelamin A accumulation was detected in mouse embryonic fibroblasts (MEFs) and kidneys. The size and weight were reduced after four weeks of age and lifespan was shortened to 20 weeks, mostly probably due to dilated cardiomyopathy and heart failure. After two months of age, progressive loss of weight, and an abnormal posture characterized by hunched position and scoliosis were observed. Mice had decreased blood glucose and serum triglyceride levels after four weeks of age, bigger hearts and kidneys, thinning of the ventricular wall and, in some cases, dilatation of both ventricles. The immunohistochemical (IHC) analysis showed muscle degeneration foci, infiltration of inflammatory cells and interstitial fibrosis. At 16 weeks, some mice were losing their fur, whiskers and eyelashes. They completely lost the subcutaneous fat layer. The epidermis and hair follicles were atrophic and an increased number of apoptotic bodies in the basal layer of the epidermis and in hair follicles was observed [89].

Another ZMPSTE24-null mouse model was created by Bergo and colleagues by excision of exon 8. Symptoms in heterozygotes were observed by 15 months of age—they were smaller than wild-type mice, appeared weak and lost hair. Homozygotes were much smaller, muscle weakness was observed by six–eight weeks of age, they developed kyphosis of the spine, lost hair and appeared malnourished. The lifespan was longer than for the previous ZMPSTE24 model—mice were dying by six–seven months of age, probably mostly due to spontaneous bone fractures without healing, resulting in inability to eat. Prelamin A accumulation was also observed in MEFs, but no pathology was detected by IHC, especially no muscle degeneration or heart pathology [90].

A ZMPSTE24-deficient model was also created using zebrafish, but no growth retardation was observed even for homozygotes, despite prelamin A accumulation [91].

Further models were focused on lamin. Yang and colleagues created a mouse model expressing progerin only by deletion of *LMNA* intron 10, the last 150 nucleotides of exon 11 and intron 11. Lamins A and C were not detected for homozygotes and were at the lower level than progerin for heterozygotes. Symptoms were also observed for heterozygotes (HG/+). By six–eight weeks they began to lose weight and they died by about 27 weeks of age. Less subcutaneous fat and abdominal fat, kyphosis of the spine, osteolytic lesions, bone abnormalities (e.g., micrognathia), and misshapen nuclei with large blebs in MEFs were observed. In contrast to ZMPSTE24 models, no changes in skeletal muscles were observed. Heterozygotes without wild-type lamin A (HG/−) had a more severe phenotype and died by 10–14 weeks of age. The most severe symptoms were observed for homozygotes (HG/HG). They were very small, had no adipose tissue, micrognathia and an abnormal skull shape with open cranial structures, spontaneous bone fractures, misshapen MEF nuclei to a greater extent and died by three–four weeks of age [34,92,93,94,95].

It should be pointed out that *LMNA*-null mice had similar lifespan. Depending on mouse model, they died before birth [96] or 2–3 weeks after births [97] however the main observed symptom was muscle dysfunction. Interestingly, mice models with only prelamin A expression or lamin A only, did not show significant phenotype, similarly to the mouse model with lamin C only [98].

Other attempts were made to create tissue-specific models. Wang and colleagues developed transgenic mice expressing wild-type lamin or progerin in epidermis only, under control of a keratin 14 promoter. The minigene was tagged with FLAG and expressed at twice the endogenous lamin level. Mice had a normal growth rate and life span, with no alterations in skin, hair follicles or hair. However, for cultured primary epidermal keratinocytes with progerin overexpression nuclear shape alterations were detected [99].

Tissue-specific progerin expression was also obtained by Sagelius and colleagues using tetracycline-inducible mouse transgenic lines that carry a minigene of human *LMNA*, restricted to tissues expressing keratin 5. Wild-type lamin and progerin minigenes consisted of exons one–11, intron 11 and exon 12. For progerin additionally the mutation c.1824C > T in exon 11 was introduced, resulting in aberrant splicing typical for HGPS. Series of mice differing in expression levels were analyzed. For some of them, hair thinning, growth retardation, premature death, abnormalities in the skin and teeth, fibrosis, and loss of hypodermal adipocytes were observed [100].

Another mice model was generated by the same team and they analyzed the effect of progerin expression in preadipose and adipose cells. In generated mice model about 5% of adipocytes from white adipose tissue (WAT) expressed progerin. In long term (up to 90 weeks) this resulted in increased proliferation of progerin-expressing preadipocytes and adipocytes, accumulation of DNA damage, infiltration of macrophages followed by increased senescence of adipose tissue, progressive lipoatrophy, fibrosis and general inflammation [101]. Thus, oxidative stress seems might be common feature in progeria since it has been also reported in patient fibroblasts [68] and provides another evidence for complexity of mechanism leading to phenotype development in HGPS. Since conserved C-terminal cysteine residues in lamin A/C were identified as prone to oxidative stress and DNA damage signaling [66,102] one might assume that different C-terminus of progerin might disrupt this signaling.

Also models that introduce a point mutation into the *LMNA* gene were developed. Mounkes and colleagues created a mouse line with the L530P mutation. In humans, presence of this substitution results in autosomal dominant EDMD, but in mouse progeria-like symptoms were observed and aberrant splicing between exons 9 and 10 was detected. Homozygotes (L530P/L530P) grew slower and died within four–five weeks. They had a waddling gait (suggesting immobility of joints), micrognathia and abnormal dentition, many skin symptoms such as hyperkeratosis or increased deposits of collagen, lack of the subcutaneous fat layer, decreased density of hair follicles, and decreased bone density. Some degeneration of skeletal muscles and heart was also observed, but without typical dystrophic markers. For MEFs, discontinuities of the nuclear envelope, lamin C in cytoplasm, and shortened lifespan were observed [103].

The first mouse model that directly reflects the most common HGPS mutation was created by Osorio and colleagues. A point mutation c.1827C > T was introduced into the *LMNA* locus using the Cre/lox system, resulting in presence of murine progerin (p.G609G) and lamin C with no lamin A. Of note, although this is not the “common” (human) c.1824C > T (p.G608G) mutation, it also results in cryptic splice site activation and progerin synthesis. Heterozygotes (p.G609G/+) had normal weight, size, and fertility until about eight months of age, but later they lost weight and died. Additionally, nuclear abnormalities were observed. Homozygotes (p.G609G/G609G) were infertile, progressively losing weight and died by three months. An abnormal posture, marked curvature of the spine, loss of the principal fat deposits and subcutaneous fat layer, attrition of hair follicles, positive beta-galactosidase staining in lung and liver, involution of thymus and spleen, reduction in bone density with increased porosity, loss of vascular smooth muscle cells in the aortic arch, bradycardia, decrease in serum levels of glucose, IGF-1, insulin, leptin, growth hormone, and adiponectin were observed. The global gene expression profile was comparable to ZMPSTE24 deficient mice. MEFs had large and abnormally shaped nuclei [13,104].

The following two models represent a different perspective on progeria. Both ideas originated at a similar time but take advantage of the biological activity of different enzymes.

Targeting ICMT, which is responsible for isoprenylcysteine methylation (in farnesylation motif), ameliorated the phenotype in a ZMPSTE24-deficiency mouse model [85]. This indicates that inhibition of ICMT might be a useful method for treatment of progeria. Another study tested compounds affecting progerin binding to lamin A in vitro and in a mouse model of progeria (G609G phenocopy mutation) [105]. Both cellular model parameters and mice life span were significantly improved.

Another compound, devised to inhibit acetyl transferase, NAT10 was successfully used in patients’ fibroblasts to treat progeria [106]. Knockdown of NAT10 level and inhibition of NAT10 activity were similarly efficient. Chemical inhibition of NAT10 as well as its knockout in a progeria mouse model (G609G phenocopy) significantly enhanced the lifespan of animals [107]. Since NAT10 exhibits not only N-acetyltransferase activity but also other activities and functions, it is difficult to discuss potential therapeutic mechanisms [108,109]. Definitively, the level of progerin and other lamins seems to be unaffected. The authors suggested a mechanism based on improved nuclear transport as a major contributory pathway [110], but there are still too many questions to be answered before we can seriously discuss potential mechanisms of NAT10 action in HGPS.

## 4. Classical Treatment Strategies for HGPS Progeria

Many small molecule drugs for therapy of progeria have been tested so far. Most of them were studied in tissue culture models from HGPS patients or MEFs from transgenic murine models. Only a few of them were tested in a mouse model system or during clinical trials. Most of the tested drugs were already studied earlier in other laminopathies and they address a particular selected signaling pathway or process. For example, to address the issue of abnormal gene regulation by progerin, a variety of histone deacetylase inhibitors such as trichostatin A, sulforaphane or butyrate have been used [81,111,112] in order to reverse the general gene repression trend. Another option was to use retinoic acid (RA) or all-trans retinoic acid (ATRA) for modification of lamin A/C level by regulation of transcription by ATRA-sensitive elements in the *LMNA* promoter [113]. In order to address cardiovascular problems, ERK1-3 kinase inhibitors were tested in laminopathies with cardiac phenotype, and they can possibly be used for amelioration of cardiovascular phenotype in progeria [114,115]. To address the issue of a high level of prenylated lamin A, farnesyl transferase inhibitors have been used together with drugs enhancing autophagy [80,116,117]. For a detailed review on the subject see: [13,118,119,120]. Here we need to mention a small molecule approach targeted to inhibit carboxyl methyltransferase responsible for methylation of the cysteine residue in lamins with a CAAX motif [85,121] and to use NAT10 inhibitors for laminopathies and specifically for progeria [106,108,109]. We discussed these issues in detail in the Animal Models section.

A separate question is how to properly define the critical disease markers and what markers can be used for proper assessment of the efficiency of a particular treatment procedure. Which set of markers is sufficient for cellular model studies and which sets are necessary for animal model studies?

For tissue culture model studies several phenotypic markers of the disease have been used. The most frequently used are: Lobulated nuclei, beta-galactosidase level, proportion of progerin (protein and transcript) to lamin A, C and B, mechanical properties of nuclei, efficiency of mitochondria, nuclear transport, proliferation rate, activity of mTOR, ERK1-3 and Akt pathways, reactive oxygen species (ROS) level, chromatin markers such as H3K9me3, HP1α, H3K27acetyl and H3K27me3, level of LAP2, or DNA repair markers. RNAseq or any other means of gene expression profiling for controls and treatment have not been frequently used, but data gathered so far from these analyses are invaluable [122,123]. Due to such variety of treatment efficiency markers it is sometimes impossible to perform comparative analyses of phenotype improvement upon treatment, especially at the level of a tissue culture model system, especially when only one or few markers are analyzed such as nuclear blebbing, progerin level or proliferation rate. Similarly, but on a smaller scale, due to the limited animal models available and studies performed, it is hard to compare the efficiency of treatment strategies in model systems.

Interestingly, HGPS patient fibroblasts with the classical mutation show characteristic lobulated nuclei. This cellular phenotype is frequently used in cellular studies and animal model studies as a readily visible, obvious marker of the disease, and the efficiency of the treatment procedures.

Nuclear envelope lobulation can be reversed in such model cells by progerin silencing, inhibition of farnesylation, induction of autophagy, or a combination of them. The problem is that lobulation disappearance is not a valid method of analyses of improvement of the mechanisms at the molecular level.

Similar lobulated nuclei can also be created by transfection of control fibroblasts with progerin protein (and frequently prelamin A or lamin B) but typically not by lamin A or lamin C proteins. It is therefore possible to create “lobulated nuclei”—a cellular model of HGPS progeria using any cell type, which is currently necessary for addressing a particular question. The major drawback of lobulated phenotype for monitoring efficiency of treatment is its superficial value. Such lobulated nuclei can also be created by overexpression of C-terminal lamin B fragments or other protein fragments containing a farnesylation motif [124]. This suggests that it is more an issue of mechanical and sterical problems for the cell nucleus than it is really a functional one. Interestingly, overexpression of lamin A in progeria fibroblasts does not improve the phenotype [125]. Clinical trials of combined treatment with small molecules performed so far [126,127,128,129] demonstrated limited but statistically significant overall improvement of patients and statistically significant prolongation of lifespan but with probably limited potential for a further increase in efficiency. The possible limitations are the therapeutic windows and dose limits for a particular drug, unknown contribution of a particular drug to the overall therapeutic effect, and possible lack of correlation or synergy between molecular effects of each drug administered at the same time point since in patient fibroblasts more efficient administration of drugs seems to be crucial [111].

## 5. Gene Therapy for HGPS Progeria

Several separate discoveries have made the consideration of gene therapy for HGPS possible. Tissue culture model studies and then animal model studies revealed that if we block progerin expression, even together with lamin A expression, it is still beneficial compared to the starting disease model [34,98,130,131,132].

The second discovery originated from the knowledge that human neuronal tissues are not affected by HGPS phenotype. The molecular background of this phenomenon was based on specific, epigenetic silencing of lamin A and progerin transcripts by microRNA. It was demonstrated that in neurons there is present microRNA-9 which has two binding sites in the 3’UTR region of the lamin A and progerin transcripts (see Figure 1) but not in the 3’UTR for the lamin C transcript. miR-9 silenced expression of lamin A and progerin protein, keeping neurons free of disease and allowing normal function [92,105,133]. The role of miR-9 and target sites in the lamin A-specific transcript was further confirmed by studies in animal models. The strongest silencing effect for lamin A transcripts was observed in the cerebral cortex and cerebellum [35,93]. This is in perfect agreement with the level of miR-9 in these tissues. Figure 2 illustrates the origin of all three different primary transcripts giving rise to the mature miR-9 sequence. Note that each of the primary miR-9 sequence can be a part of many transcription products from each particular open reading frame (ORF). Only a few such transcripts are directed by miR-9 own regulatory elements. Table 1 illustrates the relative abundance of each miR-9 primarily transcript. Data were obtained from the Fantom database and unfortunately are far from satisfactory for our considerations. This is due to the limited availability of samples. The most important for our considerations is the relative abundance of mir-9-5p (mature, final form after processing, which is the same for all three pre-miRs) in tissues. This is the end-product of microRNA processing and an active component of the silencing complex. It is readily visible that the highest abundance of active miR9-5p particles is in neuronal tissues.

Since, in neurons, there already exists a natural “gene therapy” strategy, we have a proof of concept that such a strategy for gene therapy might work.

The third important discovery in favor of gene therapy is the observation from mouse model system studies with switchable progerin expression [134]. The mouse model developed abnormalities of the skin and teeth due to the use of keratin 5 promoter for progerin expression. Four weeks after progerin expression was switched off, the phenotype started to improve and full recovery was observed after 13 weeks. The same group in another study demonstrated that complete silencing of the transgenic expression of progerin normalized bone morphology and mineralization already after seven weeks [135]. While the recovery was impressive, the phenotype remission was not perfect. These data indicate that if efficient gene therapy is developed, aimed at progerin elimination, it might be successful, provided that is administered at the optimal time point, preferably before the phenotype onset.

For the consideration of potential strategies for gene therapy for HGPS the best starting points should be the systemic in vivo strategies developed for muscular dystrophies, especially Duchenne muscular dystrophy (DMD), which is an X-linked disease. Based on the assumption that female carriers do not develop the DMD phenotype while having mosaic expression of the X chromosome during development and adult life, we know that the gene therapy need not be perfectly efficient. Secondly, tissue culture studies demonstrate the dramatic improvement of cellular markers simply after a decrease in progerin level or by shifting the ratio of progerin to lamin A/C [136]. Therefore, we might assume that, for HGPS progeria, the efficiency of therapy also need not be perfect. This assumption can be supported by studies on mosaic mouse models of progeria [137]. Mosaic mice do not develop the progeroid phenotype in tissues despite the gradual accumulation of prelamin A in cell nuclei with age.

Assuming such wide border conditions for possible gene therapy strategies, our choices of the best strategy are much wider. We may consider the following strategies for HGPS progeria: (1) Gene correction; (2) correction of aberrant splicing; (3) selective knockout/knockdown of progerin; (4) selective knockdown of lamin A and progerin (plus overexpression of wt lamin A eventually); and (5) increased transcription of wt lamin A/C allele/overexpression of lamin A/C.

Considering gene therapy, the good news is that it has been possible to cure muscular dystrophies in mice models very efficiently for almost 20 years (including DMD in an mdx model) using a large variety of strategies [138,139,140,141,142,143], also including CRISPR/Cas9 successfully performed. However, in order to succeed, hundreds of animal studies were performed, testing sometimes rather exotic strategies. The bad news is that no successful clinical trial for muscular dystrophy in humans has yet been reported.

Therefore, it is important to discuss in advance possible strategies for HGPS progeria available and already preselected in other disorders.

The gene correction method seems to be a permanent, ideal, and the most elegant strategy to aim at. It seems that this strategy works perfectly in a tissue culture model system and mouse model system. This method has been already tested with HGPS patient cells using homologous recombination [144]. Use of the CRISPR/Cas system (with self-inactivating Cas) will significantly increase efficiency and decrease side effects in vivo. Targeting this strategy at entire exon 11 replacement would also minimize the side effects, but there will still be a lot of technical problems to address, for instance how to target the mutant allele in order to increase the efficiency or how to minimize the off target modifications.

Correction of the aberrant splicing is also a method previously tested for muscular dystrophies in mouse models. Adaptation of the method for HGPS mouse models seems to be more challenging technically. The use of lentivirus or AAV delivering the expression cassette for oligo RNA fully complementary to the mutation site (c.1824C > T) might either block aberrant splicing or block translation of progerin or lamin A/C if not properly preselected for specificity and efficiency. Since this strategy is similar in design and nucleic acid drug delivery to the next strategy—selective silencing of progerin—we should discuss these strategies together. Both of them should use a well-designed specific genetic drug. At the last stage of action it should be oligo RNA specifically targeting the progerin transcript only. Obviously, since therapy should be permanent, the therapeutic particle should be permanently present in targeted cells. Therefore delivery of oligo RNA, morpholino RNA, plasmid-encoded oligo RNA or plasmid-encoded microRNA appears not to be useful in therapy of patients. It must be a virus-delivered expression cassette, preferably integrated with genomic DNA, transcriptionally active in targeted tissues.

## 6. Gene Therapy Strategies Tested so far for Progeria

As was mentioned above, HGPS, as a genetic disease, is a good candidate for the application of gene therapy as a treatment method. There are only few strategies implemented so far: specific silencing of mature transcript variant 7, which codes for progerin; blocking the cryptic splicing site of pre-mRNA of the *LMNA* gene directly or indirectly, which leads to a decrease of the level of transcript variant 7, or shifting splicing toward the lamin C instead of lamin A (or progerin) (Table 2.).

The first strategy was successfully tested in HGPS patients’ fibroblasts, bearing a c.1824C > T (p.G608G) mutation. hTERT-immortalized cells were transduced with lentiviruses coding for shRNAs. Three of them were targeted to transcript variant 7 of the *LMNA* gene (the transcript coding for progerin) and six of them to pre-mRNA to block the mutated splice site. When stable cells lines expressing shRNA were established, cells were mortalized by removing the hTERT sequence.

The best sequence (shRNA3) from all those used was the one recognizing splicing variant 7. It causes reduction of the progerin level in AG3513D and 75–8 strains to 26% or 15% compared to the control, at the same time only slightly affecting the level of lamin A. Other tested sequences were not efficient or not specific against transcript variant 7.

In fibroblasts expressing chosen shRNA fewer cells have abnormal nucleus morphology than in the control (without shRNA)—16.8% versus 60.5% in AG3513D, and 13.2% versus 49.9% in 75–8 respectively. Moreover in shRNA-expressing HGPS fibroblasts the fraction of cells expressing the senescence-associated β-galactosidase was lower than in the control, and cells with a decreased level of progerin have a higher replicative potential [146].

The same shRNA sequence, delivered by lentiviruses, was used by Liu et al. [144] and Zhang et al. [146]. Human HGPS fibroblasts AG01972, AG11498, and AG06297 were reprogrammed to induce pluripotent stem cells (iPSCs). Progerin was silenced in HGPS-iPSCs and then iPSCs were differentiated to smooth muscle cells (SMCs) or fibroblasts. It was found that in SMCs the progerin silencing causes increased proliferation potential and a smaller fraction of cells expressing the senescence-associated transcripts in comparison to controls. In HGPS-iPSC-derived fibroblasts an amelioration of nucleus morphology and restoration of the heterochromatin marker H3K9me3 were observed [144]. Moreover, in SMCs-iPSCs derived HGPS cells from donors: HGADFN167 and HGADFN164 strains in which the progerin expression was downregulated using a similar protocol, a rise of the PARP1 level was observed [146].

A different strategy was implemented by Scaffidi and Misteli—they used a morpholino oligonucleotide (Ex11) to block the cryptic splice site at pre-mRNA of the *LMNA* gene. The oligonucleotide was introduced to HGPS fibroblasts or B-lymphocytes by electroporation and a decrease of the progerin transcript level by about 90% was observed while the levels of transcripts for lamin A and C were not significantly changed (only when a high concentration of oligos was used, the levels of lamins A and C changed). Western blot analyses have also shown that in transfected HGPS cells, the level of progerin is reduced by up to 5%. In more than 90% of transfected cells, the normal morphology of nuclei was restored. Moreover, the treatment with oligonucleotides led to the restoration of normal levels of lamin B, LAP2, HP1α and also H3K9me3. Furthermore, it was observed that HGPS cells with introduced oligonucleotides restored normal mobility of lamin A in studies in which lamin fused with GFP was used. And finally, the expression level of genes which are affected in HGPS, such as matrix metalloproteinase 3 and 14 (MMP3, MMP14) or chemokine (C-C motif) ligand 8 (*CCL8*), were restored to a level similar to the control [125].

This sequence was also used in further studies. Fong et al. did not note a decreased level of transcript for progerin and protein in HGPS cells, but they suggested that it may be caused by using 2’-MOE ribose oligonucleotides instead of morpholinos or another concentration. However, they found one other sequence (ASO 365) which led to a decrease in the transcript and protein level, but only 30% [124]. Thus, it clearly indicates that not only the target sequence may be crucial but also the type of oligonucleotides and other conditions of the experiment.

In further studies in which the Ex11 sequence was used, also another sequence, Ex10, was tested. Oligonucleotide Ex10 binds to standard splice site of exon 10 and an expected, and then also observed, effect is to shift the splicing toward the lamin C and decrease the level of lamin A and progerin. Osorio et al. have shown that in HGPS fibroblasts both morpholinos cause a decrease of progerin level, but the most effective was a combination of them. They also used corresponding sequences targeting transcripts coding for progerin in mice with necessary changes in single nucleotides. Using fibroblasts from mice with the c.1827C > T mutation they observed a similar decrease in progerin level and cumulative effect of both oligos used together as in human HGPS cells. In both human and mice cells a reduction of nucleus abnormalities was noted [104]. Similar results in HGPS fibroblasts were obtained by Harhouri et al. After delivery of morpholinos Ex10 and Ex11, a decrease in the level of progerin was observed, the senescence level was lower and the fraction of cells with nucleus abnormalities was smaller [132]. Moreover, in HGPS fibroblasts the fraction of γH2AX-positive cells was similar to the control and the restoration of H3K9me3, HP1α, lamin B1, and Lap2 was observed [117].

In mice with c.1827C > T mutation that were treated with a combination of two morpholinos—Ex10 and Ex11—an increase in body weight, longer life span, decrease of degree of lordokyphosis, thicker subcutaneous fat layer and higher level of serum glucose were observed. The level of progerin was lower in such organs as the liver, kidney and heart, but was not changed in skeletal muscles. Further changes such as a thicker subcutaneous fat layer, lower intensity of staining of the senescence-associated β-galactosidase in the kidney and reduced shrinkage of the spleen and thymus were also found [104].

In our opinion, specific silencing of transcript variant 7 would be the most efficient strategy in comparison to other methods. Strategy designed and tested so far based on binding to mRNA (shRNA3) or by blocking splicing of pre-mRNA (Ex11) seem to be the most efficient and promising—they are specific to transcript variant 7 causing the significant decrease of progerin level, does not disturb the level of lamin A and C and lead to improvement of various cells’ parameters. However, shRNA3 binding to transcript variant 7 was tested only on a cell culture model, so it is not known how it would work on mice model. The other strategy based on blocking splice site was tested also on mice models giving an improvement of some mice’ parameters. On the other hand, shRNA3 was delivered by lentiviral vector, which may not be directly introduced as carrier for bigger models. As it was shown in the case of Ex11, this sequence works well if it is delivered as morpholino [125] in other form it was not efficient [124]. This different outcome might be the result of different availability in serum and a half-life of both forms of oligos, and many other issues associated with technical aspects of gene therapy. Morpholino oligos seem to have many obvious advantages over RNA-based oligos, also chemically modified in cellular model and in mice model for short term studies [147,148,149]. Their advantage might be limited though for long term studies, bigger animal studies and for clinical trials when long term effects are expected and without frequent deliveries. However, there are several methods for in vivo delivery of morpholinos and the delivery efficiency might be improved. Therefore both virus-delivered oligo and morpholinos delivery strategy are worth being tested further especially when bigger animal models and models reflecting HGPS symptoms and phenotype better than current models would be available.

## 7. Conclusions

HGPS seems to be a perfect disease for the development of gene therapy treatment due to the complex pathological mechanism which is not possible to address efficiently using a single drug or multiple drug combinations due to the multiplicity of affected pathways [3,42,53,71], as discussed in this article. The genetic background of the mutation in HGPS predisposes to gene expression elimination-directed gene therapy. Based on the experiences with a development strategy for DMD, it would be beneficial to the progeria field of research to develop a better, which means bigger, model of the disease than mouse models. The development of bigger animal models for all laminopathies would be beneficial for development of our knowledge and would speed up the pre-clinical studies. Preparation of such models for HGPS has already been started in several laboratories, although no publicly available reports have been published so far. It is essential to determine very precisely the best starting time point (age) for such therapy well in advance using pre-clinical models. It would be advisable to obtain substantial evidence of whether the developed gene therapy can be ubiquitous or needs to be tissue or organ specific—in both cases delivery has to be systemic, since this will considerably modify the technical solution options for the strategy.

## Figures and Tables

**Figure 1 cells-08-00088-f001:**
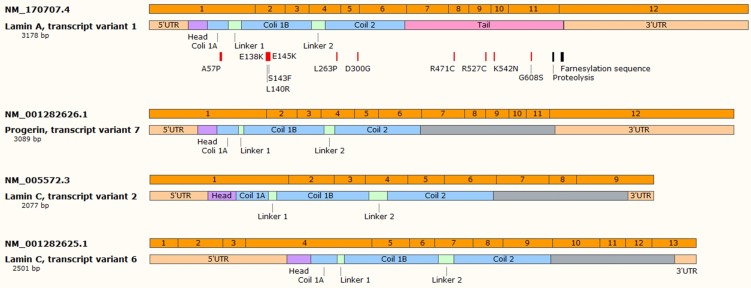
The selected transcripts of *LMNA* gene—transcript variant 1 (coding for lamin A protein) and transcript variants 2 and 6 (coding for lamin C), and transcript variant 7 arisen from mutated *LMNA* gene (coding for progerin). Farnesylated sequence and proteolytic site are indicated for lamin A transcript, as well as possible mutations in protein-coding sequence. Indicated miR-9-binding sites are present both in mRNA variant 1 and 7. siRNA binding site in present only in mRNA variant 7. Figure created based on NCBI database.

**Figure 2 cells-08-00088-f002:**
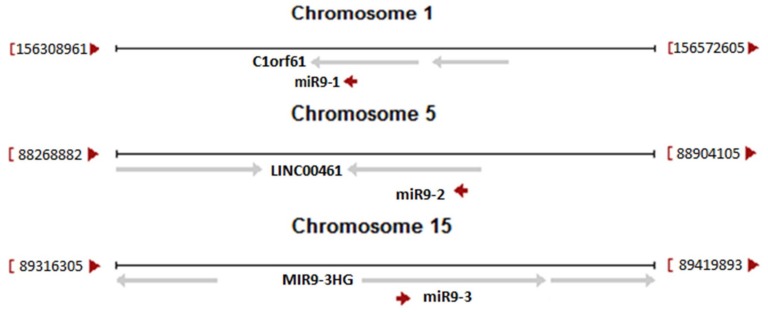
Pre-miR-9-1, pre-miR-9-2 and pre-miR-9-3 localization in human genome (red arrows). Some of host genes are also indicated (gray arrows). From C1orf61 27 transcripts could be transcribed; 6 of them are protein-coding, 15 of them contain pre-miR-9-1 sequence. LINC00461 codes 16 non-protein coding transcripts; 10 of them contain pre-miR-9-2 sequence. From MIR9-3HG 12 non-protein coding transcripts could be produced, but only 3 transcripts contain pre-miR-9-3 sequence. Figure was created based on NCBI data for listed sequences (MIR9-1 ID: 407046, MIR9-2 ID: 407047, MIR9-3 ID: 407051).

**Table 1 cells-08-00088-t001:** Table presents expression patterns and relative abundance of pre-miR-9-1, pre-miR-9-2 and pre-miR-9-3, as well as miR-9-5p, functional element of all miR-9 in RISC complex, in several human cell types. Shown data suggest that miR-9-5p is predominantly created from pre-miR-9-1 (e.g., in cerebral cortex astrocytes or in spinal cord), but pre-miR-9-2 can also predominate in some cell types. Table was created based on FANTOM5 miRNA atlas (http://fantom.gsc.riken.jp/5/suppl/De_Rie_et_al_2017/).

Probe	Pre-miR-9-1	Pre-miR-9-2	Pre-miR-9-3	miR-9-5p
Astrocyte-cerebral cortex, donor1	424	81	4	23,838
Astrocyte-cerebral cortex, donor2	56	35	1	19,110
Astrocyte-cerebral cortex, donor3	292	70	7	32,604
Astrocyte-cerebellum, donor1	13	26	0	13,025
Astrocyte-cerebellum, donor3	16	34	0	13,727
Spinal cord, adult, donor10252	66	7	1	8,708
Pineal gland, adult, donor10252	8	2	5	3,079
Fibroblast-Mammary, donor1	0	0	0	0
Fibroblast-Mammary, donor2	0	0	0	3
Fibroblast-Mammary, donor3	0	0	0	914
Prostate Epithelial Cells (polarized), donor1	0	0	0	848
Smooth Muscle Cells-Brain Vascular, donor1	0	0	0	25
Smooth Muscle Cells-Brain Vascular, donor2	0	5	0	582
Smooth Muscle Cells-Brain Vascular, donor3	0	0	0	50
Schwann Cells, donor1	0	1	0	378

**Table 2 cells-08-00088-t002:** Gene therapy tests and strategy used for HGPS treatment on tissue culture and mouse models.

Strategy	Reference	Name and Target Sequence 5′->3′	Target Transcript
Silencing of transcript variant 7, binds specifically to transcript coding for progerin	[144]	shRNA3 GGCTCAGGAGCCCAGAGCCCC	Variant 7
	[145]		
	[146]		
Blocking the activated cryptic splice site in exon 11, binds directly to the sequence of cryptic splice site	[104]	Ex11 CTCAGGAGCCCAGGTGGGTGGACCC	Mutated pre-mRNA
	[117]		
	[124]		
	[125]		
	[132]		
Blocking the activated cryptic splice site in exon 11, binds upstream of this site	[124]	ASO 365 CTGTGCGGGACCTGCGGGCA	Mutated pre-mRNA
Binding to exon 10 splice site and shift splicing towards lamin C	[104]	Ex10 CCATCACCACCACGTGAGTGGTAGC	Pre-mRNA, mutated pre-mRNA
	[117]		
	[132]		

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
