# Peer review of "Hutchinson-Gilford Progeria Syndrome—Current Status and Prospects for Gene Therapy Treatment"

_cells, 2019, doi:10.3390/cells8020088_

Round 1

Reviewer 1 Report

The review by Machowska and colleagues is a very interesting and well organized overview of current knowledge on Hutchinson-Gilford progeria syndrome, one of the most severe laminopathies.  HGPS is a paradigm of normal aging as far as metabolic and epigenetic condition of cells is concerned.  Thus, besides the interest in the pathogenetic mechanisms of that devasting rare disease affecting children, there is great engagement in HGPS research due to potential benefit of the whole field of aging research.

Major points to be addressed by the authors:

- GENERAL ASPECTS: The authors often refer to progeroid laminopathies other than HGPS and include all diseases in the same category. A clear distinction is necessary throughout the paper because both molecular defect (presence vs absence of progerin) and clinical outcome of HGPS vs other laminopathies featuring premature/accelerated  aging are different.

- FPLD should be "FPLD2" throughout the paper. There are many other FPLD forms linked to other genes.

- Some references are missing, which still provide insights into the knowledge of HGPS mechanisms. Ex: Papers showing mevinolin or FTI effects in HGPS (Columbaro et al 2005; Mallampalli et al 2005), papers showing systemic effects in progeroid mice (Osorio e at 2012; deLa Rosa et al 2013).

- ABSTRACT:  line 17, please delete "Depending on nature and location". Laminopathies do not show a clear genotype-phenotype correlation except maybe for HGPS and for the single hotspot of FPLD2. The statement cannot be considered a general rule.

- ABSTRACT: please, delete "APL...". Acquired partial lipodystrophy is linked to LMNB2 mutations, not to LMNA.

- INTRODUCTION: line 57 "among other functions". Which functions? Please also add a ref.

- PARAGRAPH: PHENOTYPE AND GENETIC BACKGROUND: "dependent on type and location of the mutation". The mechanism is more complex, also depending on patient genetic background, and yet not clear. Please, comment on this.

- PARAGRAPH: PHENOTYPE AND GENETIC BACKGROUND: "double mutation in LMNA", please specify "very rare double mutation in LMNA or ZMPSTE24 homozygous mutations "

- PARAGRAPH: PHENOTYPE AND GENETIC BACKGROUND: "less frequent phenotype of HGPS progeria". As said above, this nomenclature is misleading. Please, use the term  "other progeroid laminopathies" instead  of "HGPS progeria".

- PARAGRAPH: PHENOTYPE AND GENETIC BACKGROUND: "Indeed such pathways were discovered and..." This sentence should be tuned down, such as "a few pathways were discovered and a few treatments were proposed "

- PARAGRAPH: PHENOTYPE AND GENETIC BACKGROUND: "ERK1/3 kinase inhibitors were tested ...for progeria", This is not correct. No treatment aimed at inhibition of ERK1/3 has been tested in HGPS. Moreover the reference "Matitioli et al., 2008" is not correct and reports different data.

- PARAGRAPH: PHENOTYPE AND GENETIC BACKGROUND: line 288 "for laminopathies", which laminopathies?

- PARAGRAPH: PHENOTYPE AND GENETIC BACKGROUND: the sentence i lines 427-428 does not make sense. Please, correct or remove it.

- CONCLUSION: "which is not possible to address efficiently using single drug or multiple drug...". Please, explain the reason for this statement and mention some ref.

Author Response

Point by point Authors’ response to Reviewers Comments

First of all I would like to thank a lot Reviewers for very valuable comments giving us a chance to correct our manuscript and make it better and more interesting (hopefuly) to the potential readers.

I general we agreed to all of the comments and suggestions and tried to implement them as much as we can.  

After edition of manuscript according to Reviewers comments, manuscript went through native speaker profesional language edition.

 Please find below the list of specific actions we have taken to accomodate Reviewers suggestions: 

Reviewer No 1 comments and suggestions:

- GENERAL ASPECTS: The authors often refer to progeroid laminopathies other than HGPS and include all diseases in the same category. A clear distinction is necessary throughout the paper because both molecular defect (presence vs absence of progerin) and clinical outcome of HGPS vs other laminopathies featuring premature/accelerated  aging are different.

Yes. We agree. We introduced clear distinction between HGPS and “progeroid laminopathies” or MADA phenotype as suggested also by other reviewers since similar distinction between HGPS and atypical progeroid syndromes (or MADA) was also suggested by other reviewer (No 2). In Introduction section we defined the HGPS and progeroid laminopathie including MADA.

- FPLD should be "FPLD2" throughout the paper. There are many other FPLD forms linked to other genes.

Agreed. Done.

- Some references are missing, which still provide insights into the knowledge of HGPS mechanisms. Ex: Papers showing mevinolin or FTI effects in HGPS (Columbaro et al 2005; Mallampalli et al 2005), papers showing systemic effects in progeroid mice (Osorio e at 2012; deLa Rosa et al 2013).

We added references mentioned by Reviewer to the manuscript. The first two to the section:  “Classical treatment strategies for HGPS progeria” The last two into the animal model section.

- ABSTRACT:  line 17, please delete "Depending on nature and location". Laminopathies do not show a clear genotype-phenotype correlation except maybe for HGPS and for the single hotspot of FPLD2. The statement cannot be considered a general rule.

The sentence was removed. We added instaed a paragraph stating variability of phenotype in progeroid laminopathies using R527 residue as an example and citing proper references.

- ABSTRACT: please, delete "APL...". Acquired partial lipodystrophy is linked to LMNB2 mutations, not to LMNA.

Agreed. Done. Sorry for the mistake. We also edited entire paragraph describing laminopathies.

- INTRODUCTION: line 57 "among other functions". Which functions? Please also add a ref.

- PARAGRAPH: PHENOTYPE AND GENETIC BACKGROUND: "dependent on type and location of the mutation". The mechanism is more complex, also depending on patient genetic background, and yet not clear. Please, comment on this.

We edited the fragment accordingly.

- PARAGRAPH: PHENOTYPE AND GENETIC BACKGROUND: "double mutation in LMNA", please specify "very rare double mutation in LMNA or ZMPSTE24 homozygous mutations "

We edited entire section and description of  phenotype-genotype relation

- PARAGRAPH: PHENOTYPE AND GENETIC BACKGROUND: "less frequent phenotype of HGPS progeria". As said above, this nomenclature is misleading. Please, use the term  "other progeroid laminopathies" instead  of "HGPS progeria".

Agreed and corrected. We use name HGPS and progeroid laminopathies or MADA

- PARAGRAPH: PHENOTYPE AND GENETIC BACKGROUND: "Indeed such pathways were discovered and..." This sentence should be tuned down, such as "a few pathways were discovered and a few treatments were proposed "

Agreed and corrected as suggested.

- PARAGRAPH: PHENOTYPE AND GENETIC BACKGROUND: "ERK1/3 kinase inhibitors were tested ...for progeria", This is not correct. No treatment aimed at inhibition of ERK1/3 has been tested in HGPS. Moreover the reference "Matitioli et al., 2008" is not correct and reports different data.

Sorry for mistake this was our misexpression. Agreed and corrected the sentence abou ERK1/3 inhibitors into possibility of use to alleviate cardiovascular phenotype in progeria. New reference added.

- PARAGRAPH: PHENOTYPE AND GENETIC BACKGROUND: line 288 "for laminopathies", which laminopathies?

For laminopathies with cardiac phenotype such as EDMD. Corrected.

PARAGRAPH: PHENOTYPE AND GENETIC BACKGROUND: the sentence i lines 427-428 does not make sense. Please, correct or remove it.

Sentences corrected and section edited anyway

CONCLUSION: "which is not possible to address efficiently using single drug or multiple drug...". Please, explain the reason for this statement and mention some ref.

Agreed. Corrected.

Reviewer 2 Report

Machowska et al. aim to provide a current view and prospects for gene therapy treatment in HGPS. While this is an interesting topic there are several issues with this manuscript as it is now. First of all the manuscript needs to be proofread by a native speaker or an expert in English editing and academic writing to improve the syntax and flow of the language. I would also recommend a general restructuring of the article. Nearly two pages of describing animal models without a direct link to the main topic is going to lose the reader (at least this happened to this reviewer). This could be streamlined to less than half a page (the authors start writing about gene therapy - which is the topic of the article - on page 7 out of 12 as it is now). It might also make sense to give a more structured overview of the non-gene-therapy approaches. While there are many approaches covered it is tough reading and feels a bit like jumping between things, FTI as the one approach already tested in patients is mentioned only very briefly without details. It would make sense to break the text apart with subheadings for some topics (like genome organization, signaling…). Furthermore the text feels a bit incomplete and out of date. For example:

o   MAD is mentioned, later you talk about MADA without mentioning that this is the MAD type caused by LMNA mutations not ZMPSTE24 mutations (MADB) – this should be explained to the reader.

o   ADLD is not linked to LMNA but LMNB1.

o   AWS was used in the original publication to describe the patients, but it was published later on that this is Malouf syndrome instead.

o   “progeria has two modes of inheritance” (p2 L65) – there is only autosomal dominant inheritance for HGPS. The other mentioned mutations (p3 l103-104) are considered to be atypical progeroid syndromes or MADA, the mode of inheritance for the mentioned mutations can be recessive or dominant (dependent on the mutation). Those are having similarities to HGPS, but they are clinically different (and the effect on protein level is different as they don't affect the processing as far as we know).

o   Figure 1 isn’t giving any useful information at this point and the way it is drawn. I would suggest to redraw it focusing on LMNA and showing potential target sites for gene therapy or to drop it – as it is it doesn’t contribute to the manuscript.

o   P7 l285 “a new small molecule approach” gives two references from 2013, when this review will be online it will be 6 years since publication of those data – I would rephrase.

o   HGPS is not the most severe laminopathy (first sentence in the abstract), this fits for restrictive dermopathy instead.

As there are many points that should be addressed first to make the manuscript more readable I didn’t look at minor points like the usage of references yet as this would be redundant work.

Author Response

Point by point Authors’ response to Reviewers’ Comments

First of all I would like to thank a lot Reviewers for very valuable comments giving us a chance to correct our manuscript and make it better and more interesting (hopefuly) to the potential readers. In general we agreed to all of the comments and suggestions and tried to implement them as much as we can. 

After edition of manuscript according to Reviewers comments, manuscript went through native speaker profesional language edition.

 Please find below the list of specific actions we have taken to accomodate Reviewers suggestions: 

Please find below the list of specific actions we have taken to accomodate Reviewers suggestions: 

Reviewer No 2 comments and suggestions:

Machowska et al. aim to provide a current view and prospects for gene therapy treatment in HGPS. While this is an interesting topic there are several issues with this manuscript as it is now. First of all the manuscript needs to be proofread by a native speaker or an expert in English editing and academic writing to improve the syntax and flow of the language. I would also recommend a general restructuring of the article. Nearly two pages of describing animal models without a direct link to the main topic is going to lose the reader (at least this happened to this reviewer). This could be streamlined to less than half a page (the authors start writing about gene therapy - which is the topic of the article - on page 7 out of 12 as it is now). It might also make sense to give a more structured overview of the non-gene-therapy approaches. While there are many approaches covered it is tough reading and feels a bit like jumping between things, FTI as the one approach already tested in patients is mentioned only very briefly without details. It would make sense to break the text apart with subheadings for some topics (like genome organization, signaling…). Furthermore the text feels a bit incomplete and out of date. For example:

MAD is mentioned, later you talk about MADA without mentioning that this is the MAD type caused by LMNA mutations not ZMPSTE24 mutations (MADB) – this should be explained to the reader.

Corrected, both names explained and described as requested through entire manuscript. It was also the suggestion of Reviewer No1.

ADLD is not linked to LMNA but LMNB1.

Agreed. Deleted. Entire description of laminopathies and progeroid laminopathies  and HGPS was revised.  Sorry for the mistake.  

AWS was used in the original publication to describe the patients, but it was published later on that this is Malouf syndrome instead.

Agreed and corrected.

“progeria has two modes of inheritance” (p2 L65) – there is only autosomal dominant inheritance for HGPS. The other mentioned mutations (p3 l103-104) are considered to be atypical progeroid syndromes or MADA, the mode of inheritance for the mentioned mutations can be recessive or dominant (dependent on the mutation). Those are having similarities to HGPS, but they are clinically different (and the effect on protein level is different as they don't affect the processing as far as we know).

Corrected through entire manuscript. We use now the HGPS name and  progeroid laminopathies or MADA name with clear definition of genetic background.

Figure 1 isn’t giving any useful information at this point and the way it is drawn. I would suggest to redraw it focusing on LMNA and showing potential target sites for gene therapy or to drop it – as it is it doesn’t contribute to the manuscript.

We revised the Figure 1 by adding  target sites for miR9 and tested siRNA as requested. We provided also a more detailed description in Figure legend and referred to the figure in several sections.  

P7 l285 “a new small molecule approach” gives two references from 2013, when this review will be online it will be 6 years since publication of those data – I would rephrase.

Agreed and corrected.  By new we meant  not  new in time- wise but new as different approach.

HGPS is not the most severe laminopathy (first sentence in the abstract), this fits for restrictive dermopathy instead.

Agreed and corrected.

As there are many points that should be addressed first to make the manuscript more readable I didn’t look at minor points like the usage of references yet as this would be redundant work.

During the revision we edited also references list.

Reviewer 3 Report

In “Hutchison-Gilford Progeria Syndrome- Current View and Prostpects for Gene Therapy Treatment” Machowska et al present some background on HGPS, a discussion of its pathology, models for studying it, current treatment regimes, and the tested and potential gene therapy approaches.  There are a number of changes I would suggest to improve the overall flow of the review listed below and also it is important that the English writing and grammar be checked by a native speaker as it is very poor throughout.

The abstract lists lots of lamin-related diseases whereas based on the title it should really give more about the content with regards to HGPS.  I would recommend replacing this disease list with text more related to the title.

At the end of the second paragraph of the phenotype and genetic background section they raise the question of to what extent aspects of progerin expression are a cause or effect of ageing.  This is actually an interesting question, the more so as several people have not been able to reproduce the Scaffidi results.  Yet the topic was dropped without going further into this controversy. It would be good to elaborate more on this and why it is difficult to get a clear answer as well as to go more into clinical variability.

            In the following paragraph the link to restrictive dermopathy is not clear and the phrasing does not clearly elucidate the differences in the lamin mutational defects between the two syndromes.  Further into the section they raise the question whether progerin interacts with A or B-type lamins, but don’t present any data relating to the question when studies have been performed.  It is good to raise open questions, but it is better in a context where data of what has been done is given.

            The hypothesis in this section that having a higher proportion of the lamin A at the periphery automatically results in all lamin-associated regions moving to the periphery and being silenced is perhaps a bit too simplistic and naïve.  The concentration of dense peripheral chromatin and epigenetic silencing marks are both decreased in HGPS and so peripheral localization of chromatin is likely not increased and would not necessarily result in silencing.  Moreover, there have been reports of B-type lamins recently in the nucleoplasm that would not be affected in theory.  The idea that targeting genome and signaling changes therapeutically might not be a viable strategy on the other hand is likely correct; however, there is little actual data with references given to address the many differences observed by those investigating individual patients.  In fact, this section might also be a good opportunity to signal a rebuke to the many in the field publishing papers off of individual patient fibroblasts without testing for commonality amongst a wider range of patients and differences with controls.

The latter concept comes across better in the next section on mouse models.  It would be better to have a transition also that links the two sections conceptually.  It would also be helpful to translate some of this information into a table because it gets lost in the verbiage and separated paragraphs.  A nice table on the other hand that conveys how different each mouse line has been would be better and conveying the concept and useful to readers.  This section ends with the ICMT and NAT10 targeting.  It would perhaps be better to move these to a separate section that includes also the FTIs, RA and vitamin D approaches. However, in all cases this analysis should include qualifications and criticisms as well.  For example, it should be highlighted that pretty much all treatments correct blebs in tissue culture cells, but have not had significant benefits to patients and the difference between tissue culture systems and in situ, particularly with respect to oxygen exposure, chaperone elevation, increased DNA damage, no proper tensile cues etc should be addressed. Likewise the fact that some effects even in tissue culture were not reproducible by other labs.  Along the same lines in the next section the statement of “overall improvement of patients and “significant prolongation of lifespan” strike me as overstated.  These are extremely minor improvements whether they can use statistics to claim significance such as a very slight increase in body weight which could come from the constipation issues associated with the FTIs.  The fact is that there is really nothing as yet that makes a notable difference in improving patient lives.

The gene therapy section a lot of emphasis is placed on the miR9 abundance in brain.  It is good that the authors note the limitations on the Fantom database data, but it may also be worth noting studies that showed lamin A staining in brain since it is clearly expressed in a number of cell types in the brain and so there could be very specific and isolated effects in the brains of HGPS patients.  I am not however certain what is meant by the sentence “Since in neurons already exist natural “gene therapy” strategy…”.  Is this referring to the types used for viral oncolytic therapies?  This tiny paragraph should be elaborated and the specific natural therapies explained.  It is also important in this section to not overstate because patients and their families will also be reading this as this field has a very engaged and active network with patient involvement.  The statement that mice models have been “cured” of MDs for 20 years makes this sound like something that would already be being used in human patients, but this is not the case.  More of an elaboration of the limitations and limits of translation from mouse models to human and what exactly is meant by “cured” in a mouse model are necessary so that these people are not filled with grossly false hopes. It should also be emphasized better what is speculation and what has been tested in this section.  The following section starts already in the second paragraph with “The first strategy was successfully used by Huang…”.  This again is too promising sounding for patients and family members who are likely to read the review and should be tempered.

The conclusions points to the argument to use “bigger” animal models and this is already being done by several groups, particularly in Spain.  I do not know if they have published this yet, but if it is appropriate it would seem better to state rather that this is a direction being taken by the field and in better context of reminding the reader of the limitations of the mouse model.

Author Response

Point by point Authors’ response to Reviewers Comments

First of all I would like to thank a lot Reviewers for very valuable comments giving us a chance to correct our manuscript and make it better and more interesting (hopefuly) to the potential readers.

I general we agreed to all of the comments and suggestions and tried to implement them as much as we can. 

After edition of manuscript according to Reviewers comments, manuscript went through native speaker profesional language edition.

Please find below the list of specific actions we have taken to accomodate Reviewers suggestions: 

Reviewer No 3 comments and suggestions:

In “Hutchison-Gilford Progeria Syndrome- Current View and Prostpects for Gene Therapy Treatment” Machowska et al present some background on HGPS, a discussion of its pathology, models for studying it, current treatment regimes, and the tested and potential gene therapy approaches.  There are a number of changes I would suggest to improve the overall flow of the review listed below and also it is important that the English writing and grammar be checked by a native speaker as it is very poor throughout.

Agreed, done by native speaker and professional language editor.

The abstract lists lots of lamin-related diseases whereas based on the title it should really give more about the content with regards to HGPS.  I would recommend replacing this disease list with text more related to the title.

Agreed. Abstract edited accordingly.  

At the end of the second paragraph of the phenotype and genetic background section they raise the question of to what extent aspects of progerin expression are a cause or effect of ageing.  This is actually an interesting question, the more so as several people have not been able to reproduce the Scaffidi results.  Yet the topic was dropped without going further into this controversy. It would be good to elaborate more on this and why it is difficult to get a clear answer as well as to go more into clinical variability.

We fully agree to the Reviewers suggestions yet we are not in the position of knowledge of failed attempts to reproduce the Scaffidi experiments. Therefore we cannot comment on that particular issue since we do not have any written reports or written confirmation of such issue . As far as we know there is only one conference report available in databases confirming activation of aberrant LMNA splicing in elderly humans but without abstract.

We decided to revise the section to accommodate other controversies related to the contradictory reports from tissue culture model and mice models regarding the toxicity of progerin or progerin farnezylation. We also attempt to discuss in following sections another controversies connected to phenotype markers suitable for treatment efficiency.

            In the following paragraph the link to restrictive dermopathy is not clear and the phrasing does not clearly elucidate the differences in the lamin mutational defects between the two syndromes.  Further into the section they raise the question whether progerin interacts with A or B-type lamins, but don’t present any data relating to the question when studies have been performed.  It is good to raise open questions, but it is better in a context where data of what has been done is given.

Agreed and corrected. The question of clear distinction and definition of HGPS and progeroid laminopathies (or MADA) was raised by Reviewer 1 and 2. 

We edited entire section and extended the section dedicated to discussion of  question of interaction of progerin with lamin A or B-type. We added the references demonstrating the data supporting both sides of the discussion and added our comments and suggestions to the ongoing discussion.

            The hypothesis in this section that having a higher proportion of the lamin A at the periphery automatically results in all lamin-associated regions moving to the periphery and being silenced is perhaps a bit too simplistic and naïve.  The concentration of dense peripheral chromatin and epigenetic silencing marks are both decreased in HGPS and so peripheral localization of chromatin is likely not increased and would not necessarily result in silencing.  Moreover, there have been reports of B-type lamins recently in the nucleoplasm that would not be affected in theory.  The idea that targeting genome and signaling changes therapeutically might not be a viable strategy on the other hand is likely correct; however, there is little actual data with references given to address the many differences observed by those investigating individual patients.  In fact, this section might also be a good opportunity to signal a rebuke to the many in the field publishing papers off of individual patient fibroblasts without testing for commonality amongst a wider range of patients and differences with controls.  

We are aware of the  drawbacks of current knowledge and controversies associated with this subject. Our example of the mechanism was described just as an example of one mechanism which can give rise to unpredicted outcomes (both ways transcriptionally-wise for a single TAD).

As for the progeroid laminopathies and chromatin we decided to revise entire section and discuss the current controversies regarding the effect of lamin A mutations on chromatin. We can find reports documenting  increase and decrease of heterochromatin markers in progeroid laminopathies as association of transcriptionally active transcriptional domains. We discussed both theories from our point of view and taking into account thoroughness of experimental procedures used.

As to the question of B-type lamins in nucleoplasm this might be an interesting issue by itself. It may participate in similar functions on chromatin as intranuclear lamin A assuming its properly phosphorylated or participate in other function. Anyway any interphase/nongerminal lamin exist in equilibrium between polymerized (supposedly lamina-associated) state and oligomeric state. Although it is very intriguing question in our opinion this is not the issue for this particular manuscript to address.

The latter concept comes across better in the next section on mouse models.  It would be better to have a transition also that links the two sections conceptually.  It would also be helpful to translate some of this information into a table because it gets lost in the verbiage and separated paragraphs.  A nice table on the other hand that conveys how different each mouse line has been would be better and conveying the concept and useful to readers.  This section ends with the ICMT and NAT10 targeting.  It would perhaps be better to move these to a separate section that includes also the FTIs, RA and vitamin D approaches. However, in all cases this analysis should include qualifications and criticisms as well.  For example, it should be highlighted that pretty much all treatments correct blebs in tissue culture cells, but have not had significant benefits to patients and the difference between tissue culture systems and in situ, particularly with respect to oxygen exposure, chaperone elevation, increased DNA damage, no proper tensile cues etc should be addressed. Likewise the fact that some effects even in tissue culture were not reproducible by other labs.  Along the same lines in the next section the statement of “overall improvement of patients and “significant prolongation of lifespan” strike me as overstated.  These are extremely minor improvements whether they can use statistics to claim significance such as a very slight increase in body weight which could come from the constipation issues associated with the FTIs.  The fact is that there is really nothing as yet that makes a notable difference in improving patient lives.

Thank you very much for the suggestions. I am afraid that in the case of several  commentsand suggestions  we are not in the position to address them in our review. We are not a clinitians but molecular biologists and we do nor feel we are the experts to discuss in details the treatment strategies in pre-clinical models and for patients. We do not focus in our review on “classical” treatment strategies, outcomes, standards used and benefits for patients. Besides during last few years several reviews dedicated to HGPS progeria have been published by expert clinicists. Some of them focused on treatment of HGPS. We do not want to be redundant.

Nevertheless, as a molecular biologists we try to discuss some of the molecular mechanisms which are  or might be addressed by particular compound or drug.

Therefore we revised the section as the previous and following one in order to accommodate  some of your suggestions. At least those we feel we have expertise at.

The gene therapy section a lot of emphasis is placed on the miR9 abundance in brain.  It is good that the authors note the limitations on the Fantom database data, but it may also be worth noting studies that showed lamin A staining in brain since it is clearly expressed in a number of cell types in the brain and so there could be very specific and isolated effects in the brains of HGPS patients.  

Agreed and corrected accordingly.

I am not however certain what is meant by the sentence “Since in neurons already exist natural “gene therapy” strategy…”.  Is this referring to the types used for viral oncolytic therapies?  

We are referring to silencing of lamin A and progerin by miR9. We edited the section to be more understandable.

This tiny paragraph should be elaborated and the specific natural therapies explained.  It is also important in this section to not overstate because patients and their families will also be reading this as this field has a very engaged and active network with patient involvement.  The statement that mice models have been “cured” of MDs for 20 years makes this sound like something that would already be being used in human patients, but this is not the case. 

We agree we should be careful. The sentence in question is actually a warning that it is not an easy to transfer the successful treatment of mice models into the clinical practice since for DMD we are waiting for almost 20 years since first mouse gene therapy and we still lack approved gene therapy for patients. Although many clinical trials (FDA approved)  have been performed so far.

 More of an elaboration of the limitations and limits of translation from mouse models to human and what exactly is meant by “cured” in a mouse model are necessary so that these people are not filled with grossly false hopes. It should also be emphasized better what is speculation and what has been tested in this section.  The following section starts already in the second paragraph with “The first strategy was successfully used by Huang…”.  This again is too promising sounding for patients and family members who are likely to read the review and should be tempered.

Agreed and taken into the considerations while editing. We discussed the problems with development and testing an efficient gebe therapy. We also added a Table 2 demonstrated the gene therapy strategies tested so far.

The conclusions points to the argument to use “bigger” animal models and this is already being done by several groups, particularly in Spain.  I do not know if they have published this yet, but if it is appropriate it would seem better to state rather that this is a direction being taken by the field and in better context of reminding the reader of the limitations of the mouse model.

We are aware of attempts to create and characterize the bigger models for HGPS. Unfortunately no public reports or conference reports have been published yet. Since I did not get a permission to report any of the attempts in this direction I cannot write about them.

We edited the conclusion section according to you suggestion mentioning that such studies are being made currently.

Round 2

Reviewer 2 Report

The quality of this review significantly increased after revision.

I have only one point that could be included:

- It might make sense to discuss following paper when discussing the expression levels of progerin/lamin A and their significance.

Revechon, G., Viceconte, N., McKenna, T., Sola Carvajal, A., Vrtacnik, P., Stenvinkel, P., et al. (2017). Rare progerin-expressing preadipocytes and adipocytes contribute to tissue depletion over time. Sci Rep, 7(1), 017-04492.

-

Author Response

I would like to thank all Reviewers for interesting and valuable comments.

We agreed to the Reviewer suggestion to discuss recommended paper (Revchon et al 2017). We cited the paper and discussed it in the context of general  phenotype development of HGPS in respect to ROS activation, oxidative stress and DNA repair and tissue atrophy. 

We also updated the section dedicated to the potential impact of lamins and progerin presence on, LADs and TADs citing and discussing recent paper by Zhang et al 2018.

Reviewer 3 Report

The manuscript has been adequately improved for publication.

Author Response

I would like to thank Reviewer again for the comments and suggestions for the manuscript.

Just to inform we added and discussed new citation (Revchon et al 2017) in "Animal model section" as suggested by other reviewer.

We also updated the discussion on the subject of potential effect of lamins and progerin on LADs and TADs structure and gene expression. We cited new paper  by Zheng et al 2018 and discussed it in the context of effect of lamins, through LADs rearrangements on TADs and gene expression profile.